# Modeling and Optimization of Laser Cladding Fixation Process for Optical Fiber Sensors in Harsh Environments

**DOI:** 10.3390/s22072569

**Published:** 2022-03-27

**Authors:** Caixia Yang, Yuegang Tan, Yi Liu, Ping Xia, Yinghao Cui, Bo Zheng

**Affiliations:** 1School of Mechanical and Electronic Engineering, Wuhan University of Technology, Wuhan 430070, China; yangcaixialy@126.com (C.Y.); ygtan@whut.edu.cn (Y.T.); xiaping525@163.com (P.X.); 2School of Mechatronics Engineering, Zhongyuan University of Technology, Zhengzhou 450007, China; yh_cui@zut.edu.cn (Y.C.); zhengbo0371@foxmail.com (B.Z.)

**Keywords:** fiber optic sensor, laser cladding, powder concentration, energy attenuation, numerical simulation

## Abstract

In order to overcome the shortcomings of the poor shear resistance of the bare optical fiber whose coating layer falls off due to harsh conditions, such as on aero-engines and the marine environment, the coaxial powder feeding laser cladding method (CPFLCM) is proposed to connect the optical fiber sensor and the substrate. The concentration field model of the powder flow is established in sections, the effective number model of particles and the corresponding laser attenuation rate are obtained. Through simulation, the influence of relevant parameters of laser cladding on the temperature field was analyzed, and the accurate parameters of laser cladding were optimized. Finally, the temperature rise trajectory of the substrate temperature field was verified by using the fiber grating temperature sensor. Through experiments, the quality of the molten pool and the optical transmission loss of the optical fiber sensor were analyzed, and the consistency of the simulation optimization parameters was verified. Through this paper, it can be concluded that the proposed CPFLCM can realize the effective connection of the optical fiber sensor to the substrate. It is of great significance in the application of optical fiber sensors in harsh environments of oceans and aerospace.

## 1. Introduction

Optical fiber sensor technology meets the development needs of modern sensor technology due to its small size, light weight, no electromagnetic interference, high measurement accuracy, high temperature resistance, and corrosion resistance. It has great application prospects in high temperature conditions such as aerospace and nuclear fusion reactors, as well as in complex and harsh marine environments. However, the coating layer of the optical fiber sensor is very easy to fall off in harsh environments, such as the high temperature environment on aero-engines and the changing and complex environment of the ocean. The detached bare optical fiber has the disadvantages of brittleness and poor shear resistance, so it is urgent to solve the fixed problem of the optical fiber sensor in this harsh environment. At present, the common fixation method for optical fiber sensors is adhesive technology with glue [1,2,3]. However, glue is limited by the material, and is easy to corrode and fall off in harsh environments, resulting in the difference between the strain and other parameters received by the fiber optic sensor and the actual parameter of the measured substrate, which affects the test accuracy [4,5]. Laser cladding technology is one of the advanced surface modification technologies in recent years [6]. It uses a high-energy laser source to melt the metal-based powder onto the metal substrate. It has the characteristics of high input energy, low distortion, and good phase change quality. Grandal et al. [7] proposed the use of laser cladding technology to embed a metal-coated optical fiber sensor with a grating into a test point to minimize the thermal and mechanical stress. Havermann et al. [8] proposed to embed FBG into SS316 components by using powder bed-based selective laser cladding (SLM). However, these methods must use pre-spread powder to cover the optical fiber sensors. The powder utilization rate is not high and the redundant powder will pollute the optical fiber sensors, especially the F-P sensors. The coaxial powder feeding laser cladding method [9] (CPFLCM) is a new choice to fix the sensors on the metal substrates. Since the powder is preheated, the cladding layer has a good consistency, which can be used to fix and protect the optical fiber sensors. In the coaxial powder feeding laser cladding process, the metal powder and inert shielding gas are output coaxially from the powder nozzle. Before the powder particles enter the molten pool, there is a short-term interaction between the powder particles and the high-energy laser beam, which causes the laser energy to attenuate [10,11], and makes the temperature of the powder particles non-linearly increase. After a part of the laser energy is absorbed by the powder, the attenuated energy reaches the surface of the substrate and melts the fallen powder particles. Under the control of the CNC machining center, it moves on the surface of the workpiece at a certain speed, and then rapidly solidifies on the substrate surface to form a cladding layer after cooling. At present, many scholars have studied the temperature field of coaxial powder feeding laser cladding. According to the Lambert Beer theorem and Mie’s theory, some scholars [12,13,14] calculated the attenuation of laser power by powder flow, and obtained the laser intensity distribution at the laser spot on the workpiece surface. The above model takes the powder flow as a whole to analyze the attenuation of laser energy and the temperature field during the laser cladding process. Wang et al. [15] established a laser energy attenuation model by introducing the concept of an effective number of powder particles. This model can realize accurate calculation of laser energy attenuation in the powder particles with any volume fraction, which has certain referential significance. The above papers only analyze and study the temperature field of the cladding process itself. The purpose of this article is to realize the effective connection between the optical fiber and the substrate. That is, to study the connection between the substrate and the metal powder and the optical fiber. At the same time, in order to ensure the connection quality and the light transmission of the optical fiber sensor, it is necessary to solve the precise parameters of the cladding process. There is little literature to study this.

This paper proposes the coaxial powder feeding laser cladding method to realize the effective and reliable fixation and protect the fiber optic sensors on the metal substrate. By modeling and analyzing the temperature fields in the cladding process, the cladding parameters are optimized for the fixation of the optical fiber sensor and to minimize the loss of the fiber. The experiment results show that the optical fiber sensors fixed by the CPFLCM have very low transmission loss and high reliability on the substrate.

## 2. Theory and Computational Model of Laser Cladding Fixation for Optical Fiber Sensors

### 2.1. Powder Flow Concentration Modeling

A four-axis coaxial powder feeding laser cladding device is used. The dimensional diagram of the laser cladding structure for optical fiber sensor fixation as shown in Figure 1a. The laser beam and the powder feeder move synchronously. After the metal powder is ejected from the nozzle, it travels a certain distance to the laser region, and is exposed to the radiation of the laser. The interaction between powder and laser beam induces two processes. Firstly, the powder is heated by the laser radiation, and the temperature of the powder rises. When the laser power is higher than the melting point, the powder begins to melt during the falling; secondly, due to the shielding and scattering effects of the powder on the laser light, the energy of the laser is attenuated when it reaches the substrate. The powder feeder sprays the powder stream at a certain speed under the protection of the protective gas, and enters the laser molten pool composed of the substrate and the optical fiber sensor. When the laser power is sufficient to melt the substrate and the powder, a stable cladding layer with good adhesion to the coated optical fiber sensor is formed. After the powder flow enters the cladding area, it first forms an annular suspended area and then forms a Gaussian distribution [16,17]. For powder particles, the real effective working area after entering the laser cladding area is shown in Figure 1b.

The interaction of the laser beam, powder flow, bare fiber, and substrate in laser cladding is complex, and there are many influencing factors. In order to establish a mathematical model describing the entire process, this article makes the following assumptions during the laser cladding process:(1)The speed parameters of the powder particles and the carrier gas leaving the nozzle have reached equilibrium, and the solid powder particles are not affected by gravity and inter-particle drag forces during the falling process;(2)The powder particles in the carrier air stream sprayed from the nozzle are spherical with uniform volume distribution;(3)Because the volume fraction of powder particles in the gas–solid two-phase flow is very low, the reflection and refraction of the laser, the mutual heating between the particles and the convection heat transfer between the particles are ignored; at the same time, it is assumed that there is no mutual shielding effect between the solid particles during the falling process of the particles;(4)The energy distribution of the laser beam is uniform.

The laser cladding structure of optical fiber sensor is shown in Figure 1a, where the cross section of the nozzle and the center of the laser beam are taken as the origin of the coordinates. The horizontal cross-section of the powder nozzle is z=0. The powder flows from the nozzle to the substrate in the vertical direction along the z axis at a constant speed. As shown in Figure 1b, the powder flow distribution is divided into four stages after the powder leaves the nozzle. Along the z axis the powder flow concentration expression is as follows:(1)$$C\left(z\right)=\{\begin{array}{l} C1\left(z\right)=\frac{Mp\;\times \;{r_{l}}^{2}}{Q\cdot \{{\left(L-\frac{z}{\tan\alpha }+\frac{2r_{l}}{\sin\alpha }\right)}^{2}-{\left(L-\frac{z}{\tan\alpha }\right)}^{2}\}}\;\;\;\;\;\;\;\;\;\;\;\;\;\;\;\;0\leq z\leq d1\\ C2\left(z\right)=\frac{Mp\;\times \;{r_{l}}^{2}}{Q\cdot \{{\left(L-\frac{z}{\tan\alpha }+\frac{2r_{l}}{\sin\alpha }\right)}^{2}-{\left(L-\frac{z}{\tan\alpha }\right)}^{2}\}}\;\;\;\;\;\;\;\;\;\;\;\;\;\;\;d1\leq z\leq d2\\ C3\left(z\right)=\frac{Mp\;\times \;{r_{l}}^{2}}{Q\cdot {\left(\frac{2r_{l}}{\sin\alpha }-\frac{z}{\tan\alpha }+L\right)}^{2}}\;\;\;\;\;\;\;\;\;\;\;\;\;\;\;\;\;\;\;\;\;\;\;\;\;\;\;\;\;\;\;\;\;\;\;\;\;\;d2\leq z\leq d3\\ C4\left(z\right)=\frac{Mp\;\times \;{r_{l}}^{2}}{Q\cdot {\left(\frac{z}{\tan\alpha }-L\right)}^{2}}\;\;\;\;\;\;\;\;\;\;\;\;\;\;\;\;\;\;\;\;\;\;\;\;\;\;\;\;\;\;\;\;\;\;\;\;\;\;\;\;\;\;\;\;\;\;d3\leq z\leq d4 \end{array}\;$$

### 2.2. The Laser Energy Attenuation Rate β

In practice, the powder enters the laser beam after leaving the nozzle and the effective volume of the laser beam is divided into four stages. The effective volume is expressed as:(2)$$S=\{\begin{array}{l} S1=0\;\;\;\;\;\;\;\;\;\;\;\;\;\;\;\;\;\;\;\;\;\;\;\;\;\;\;\;\;\;\;\;\;\;\;\;\;\;\;\;\;\;\;\;\;\;\;\;\;\;\;\;\;\;0\leq z\leq d1\\ S2=\int_{d_{1}}^{d_{2}}\pi {\left(\frac{w_{B}}{2}-L+\frac{z}{\tan\alpha }\right)}^{2}dz\;\;\;\;\;\;\;\;d1\leq z\leq d2\\ S3=\int_{d_{2}}^{d_{3}}\pi {\left(\frac{w_{B}}{2}\right)}^{2}dz\;\;\;\;\;\;\;\;\;\;\;\;\;\;\;\;\;\;\;\;\;\;\;\;\;\;\;\;d2\leq z\leq d3\\ S4=\int_{d_{3}}^{d_{4}}\pi {\left(\frac{w_{B}}{2}\right)}^{2}dz\;\;\;\;\;\;\;\;\;\;\;\;\;\;\;\;\;\;\;\;\;\;\;\;\;\;\;\;\;\;\;\;d3\leq z\leq d4 \end{array}$$

The volume of a single powder particle is $\;V_{P}=\frac{4}{3}\rho_{p}\pi {r_{p}}^{3}$. Within z∈{0,d4}, the number of powder particles that can effectively enter the laser beam is calculated as follows:(3)$$N=\frac{CS}{V_{P}}\;\;\;$$

The ability of the powder particles to absorb laser energy after the laser beam passes through the powder particles is β, which is also the laser energy attenuation rate, and can be calculated:(4)$$\beta =\frac{{r_{p}}^{2}\cdot \sum_{n=1}^{4}Cn(z)\cdot Sn}{V_{p}(\frac{w_{B}}{2}{)}^{2}}$$
where the denominator expression in the Formula (4) is the projected area of the laser spot on the substrate, and its molecular expression is the sum of the area of the effective powder particles projected to the substrate in the laser area.

### 2.3. The Temperature T of a Single Particle

The laser energy absorbed by a single particle in the laser beam area can be calculated as follows:(5)$$e_{0}=2\pi {r_{p}}^{2}q\eta t_{0}$$
where q is based on assumption (5), which can be obtained by the formula $q=\frac{4P}{\pi {w_{B}}^{2}}$; meanwhile, the effective time $t_{0}$ for a single powder particle to pass through the laser beam can be calculated: $\;t_{0}=\frac{w_{B}\pi {r_{l}}^{2}}{Q\cos\alpha }$.

When the powder particles reach the melting point temperature, the energy absorbed can be calculated:(6)$$e_{1}=\frac{4}{3}\pi {r_{p}}^{3}\rho_{p}c_{p}(T_{m}-T_{0})\;$$

Considering the latent heat of the particles, when the powder particles are completely melted, the energy absorbed can be calculated:(7)$$e_{2}=\frac{4}{3}\pi {r_{p}}^{3}\rho_{p}c_{p}(T_{m}-T_{0})+\frac{4}{3}\pi {r_{p}}^{3}\rho_{p}l_{p}$$

Then, the temperature of a single particle in the region passing through the laser beam can be calculated:(8)$$T_{\;}=\{\begin{array}{c} T_{0}+\frac{6P\eta {r_{l}}^{2}}{Qw_{B}r_{p}\rho_{p}c_{p}\cos\alpha }\;\;\;\;\;\;\;\;\;\;\;\;\;\;\;\;\;\;\;\;\;e_{0}<e_{1}\\ \;\;\;\;\;\;\;\;\;\;\;\;\;\;\;\;T_{m}\;\;\;\;\;\;\;\;\;\;\;\;\;\;\;\;\;\;\;\;\;\;\;\;\;\;\;\;\;\;\;\;\;\;\;\;\;e_{1}\leq \;e_{0}\leq e_{2}\;\\ T_{0}+\frac{6P\eta {r_{l}}^{2}}{Qw_{B}r_{p}\rho_{p}c_{p}\cos\alpha }-\frac{l_{p}}{c_{p}}\;\;\;\;\;\;\;\;\;e_{2}<e_{0} \end{array}$$

## 3. Simulation and Analysis of Energy Attenuation Model of Laser Cladding Fixation for Optical Fiber Sensors

The relationship between the relevant parameters of the laser cladding and the attenuation rate of the laser beam is calculated using MTLAB based on the model of energy attenuation as shown in Formula (4). The material properties used in the simulation are summarized in Table 1.

The simulation analysis result is shown in Figure 2. Since the laser beam spot diameter $w_{B}$ is constant in the manuscript, according to the assumption (4), the heat flux density $q=\frac{4P}{\pi {w_{B}}^{2}}$. Hence, the laser power is used instead of the laser heat flux as the laser process parameter. For powder particles, the higher the laser attenuation rate, the more laser energy the powder particles can obtain, which is conducive to the metallurgic fusion of the powder particles and the molten pool. In view of the assumption (3), there is no shielding effect of the falling powder particles. According to the two-phase kinetic theory, the effective shielding area of the powder particles in the laser beam should be no more than 5%. As shown in Figure 2, when the carrier gas flow rate is greater than 3 L/min, the attenuation rate drops sharply. In order to ensure the largest possible laser absorption rate, the carrier gas flow rate is controlled at 3~5 L/min. Based on the control of the carrier gas flow, firstly the relationship between the powder feeding speed $M_{P}\;$ and the laser attenuation rate β is analyzed from Figure 2a. It can be seen that 15 mg/min and 18 mg/min can be selected. In order to save powder, the optimal powder feeding speed $\;M_{P}\;$ is 15 mg/min. Secondly, from the analysis of the relationship between the powder particle diameter $\;2r_{p}\;$ and the laser attenuation rate β  from Figure 2b, it can be seen that the particle diameter can be selected as 75 um and 61 um, because the effect of gravity cannot be ignored if the powder particle size is too small. According to assumption (1), the optimal choice of the powder particle diameter is 75 um. Thirdly, Figure 2c,d analyzes the relationship between Nozzle exit radius $\;r_{l\;}$ and Powder nozzle-to-center distance L and the laser attenuation rate β. It can be seen that due to the size limitation of the laser nozzle, the best choices of Nozzle exit radius and Powder nozzle-to-center distance are 3 mm and 4 mm, respectively. Finally, the influence of the angle *α* between the *x*-axis and the nozzle center on the laser attenuation rate β is as shown in Figure 2e, and 57.5° is the relatively optimal solution of *α*. Through the above analysis, the selection of each parameter of the numerical analysis is shown in Table 2.

According to the Formula (8), Figure 3 shows the relationship curve between the movement time of powder particles under the laser beam and their temperature. Obviously, the longer the action time, the higher the temperature of the powder particles. The slope of the temperature is increased as the action time and the laser power is increased. In this paper, considering that the substrate is nickel alloy, the nickel powder is adopted. In Figure 3, a line with the melting point of nickel powder $T_{m}$ = 1453 °C is drawn. Below this line, the powder particles are not melted. Conversely, the powder particles begin to melt. According to Formula (7), under the parameters of Table 3 ($T_{0}$ = 25 °C, P = 100~800 W, $c_{\rho }$ = 0.46 kJ/(kg·°C), $\rho_{p}$ = 8902 kg/m^3^), the effective action time of powder particles through the laser beam is 0.0023 s, and the maximum temperature of nickel powder particles is 896.80 °C, which obviously cannot be melted. However, in order to reduce the damage to the optical fiber during metallurgical fusion, the powder particle temperature must be as close to the substrate temperature as possible. That is, the laser power should be as large as possible.

The substrate material is DZ406(Ni 69%), whose melting point is 1360 °C. After the laser beam passes through the powder particles, the remaining energy and heat flux density of the laser heat source can be calculated:(9)$$q_{s}=q(1-\beta )\;\;$$

The thermophysical parameters, specific heat and thermal conductivity of the material DZ406, are set, which are shown in Figure 4. At the same time, the density of the substrate material is input as 8965 kg/m^3^, and the initial simulation temperature as 25 °C. Furthermore, by loading boundary conditions such as effective heat flow source and convection coefficient of the substrate material, the temperature cure of the substrate is obtained as shown in Figure 5.

In Figure 5, the temperature of the substrate increases with the increase in power, and then rises slowly after the temperature rises rapidly within 0.005 s. In this paper, the laser scanning rate is set to 11 mm/s, the working distance of scanning the fiber on the substrate is a laser beam spot diameter, and the required time is about 0.27 s. Correspondingly, the melting point of the substrate can be reached only when the power is higher than 600 W, and the melting point of the powder particles can be reached only when the power is higher than 700 W. In addition, the melting point temperature of the ordinary quartz fiber is about 1723 °C. First of all, in order to ensure a better metallurgical combination of the powder and the substrate, it is necessary to ensure that the difference between the particle temperature and the substrate temperature is minimized. Furthermore, the temperature of the substrate cannot exceed the melting temperature of the optical fiber, to ensure the optical transparency of the optical fiber. In view of this, combined with the numerical simulation results of the temperature of a single particle, the optimal power parameter is 700 W.

## 4. Experimental Methods and Materials

### 4.1. Experiment Verification

The verification setup of the substrate temperature field FBG sensor is shown in Figure 6a. The experimental system consisted of the laser cladding equipment (LDF4000-100, Laserline, Koblenz, Germany), FBG demodulator (OP-FBG2000, Ouguang Technology, Wuhan, China) with the resolution of 0.1 pm and the Scan frequency of 2000 Hz. The fiber grating temperature sensor (FsFBG, FemtoFiberTec, Goslar, Germany) was placed on the substrate of the DZ406 (Ni 69%) superalloy metal plate, and the powder Nickel particles were ejected from the laser nozzle for cladding under the action of the laser beam. The substrate temperature was sensed by the sensor and demodulated by the demodulator. In order not to damage the fiber grating sensor, experiments were carried out under the power of 100 W and 200 W. In the heating curve of the substrate, the comparison results between the experiment and the simulation are shown in Figure 6b. It shows that the two curve trends are consistent when the substrate temperature reaches a steady state. However, when the substrate heats up rapidly in a transient state, the sampling frequency of the demodulator cannot meet the data collection at the heating point, which will cause certain fluctuations in the experimental data. The simulation results meet the actual working conditions of the actual substrate temperature rise.

### 4.2. Experiment and Parameter Optimization

The CPFLCM was used to realize the effective connection structure between the optical fiber sensor and the melt substrate part, as shown in Figure 7. The laser cladding equipment (LDF4000-100, Laserline, Koblenz, Germany) was adopted. The metal substrate of the DZ406 superalloy metal plate was basically fixed to the bare fiber (G652D, YOFC, Wuhan, China) with the coating layer removed. The light source (Nokoxin, Shenzhen, China) with the wavelength of 1550 nm and the optical power meter (Nokoxin, Shenzhen, China) with the wavelength of 1550 nm were arranged at both ends of the fiber. The cladding was completed by setting the relevant parameters of the laser cladding. The optical power loss of the fiber optic sensor and the surface quality of the molten pool are shown in Table 3 below.

Through analysis of Table 3, the optical transmission loss of the fiber increases with the increase in the power of powder feeding speed. The molten pool can only be formed when the laser power reaches 500 W or more. When the laser power exceeds 800 W, the fiber is damaged and the optical signal is interrupted due to the high temperature of the molten pool. However, in the laser power range of 600–700 W, the powder particles produce a molten pool on the s substrate after passing through the laser beam. When the laser power is 700 W, the molten pool surface is better and the optical transmission loss is relatively small. At the same power of 700 W, when the powder feeding rate is 12 mg/min, the powder particle concentration is so small that the powder reaching the substrate is not enough to form a continuous molten pool surface. When the powder feeding rate is 18 mg/min, due to the higher powder concentration, more powder is wasted; furthermore, the powder feeding rate is too large, the laser attenuation rate is larger, so that less energy reaches the substrate. Eventually, it leads to poor quality of the molten pool after laser cladding. Through the above analysis, under other conditions of the same parameters, the laser power is 700 W, the powder feeding speed is 15 mg/min, and the optical transmission loss of the optical fiber after laser cladding is the lowest. This is consistent with the optimization results of the previous numerical analysis. In future research, it is necessary to further analyze the metallographic structure of the molten pool, the influencing factors on the compactness of the molten pool, and the long-term reliability application research under this fixing method.

## 5. Conclusions

In view of the shortcomings of the poor shear resistance of optical fiber sensors in harsh environments, this paper proposed the CPFLCM, melting the optical fiber on the substrate. A model of the density field of the powder flow from the nozzle, the number of effective particles passing through the laser beam, and the model of the laser energy attenuation rate are established. Through simulation, the influence of relevant parameters of laser cladding on the temperature field was analyzed, and the accurate parameters of laser cladding were optimized. In order to verify the correctness of the model experimentally, a fiber grating FBG temperature sensor is proposed to verify the laser energy attenuation model. Under the guidance of the simulation results, the laser cladding experimental structure of the optical fiber sensor was built for experiments. Its results are consistent with the simulation, and show that the non-gelling and reliable adhesion between the optical fiber and the substrate was realized. It is of great significance for the fixation of optical fiber sensors in harsh environments.

## Figures and Tables

**Figure 1 sensors-22-02569-f001:**
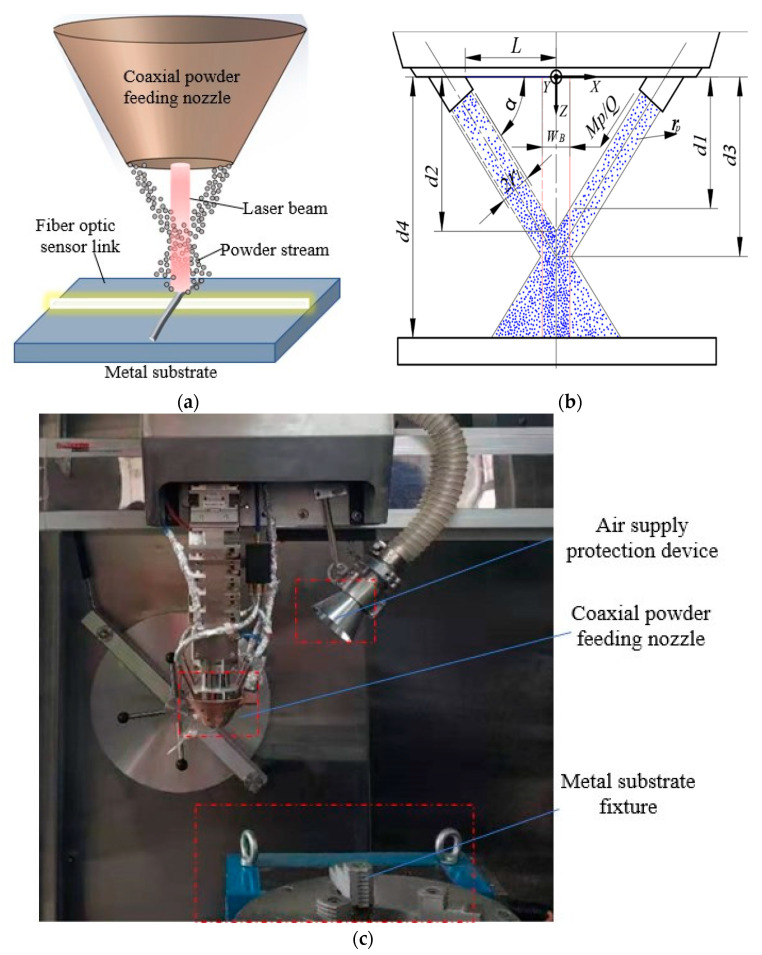
Structure diagram of laser cladding structure for optical fiber sensor: (**a**) Dimensional drawing, (**b**) Schematic diagram, (**c**) Physical image.

**Figure 2 sensors-22-02569-f002:**
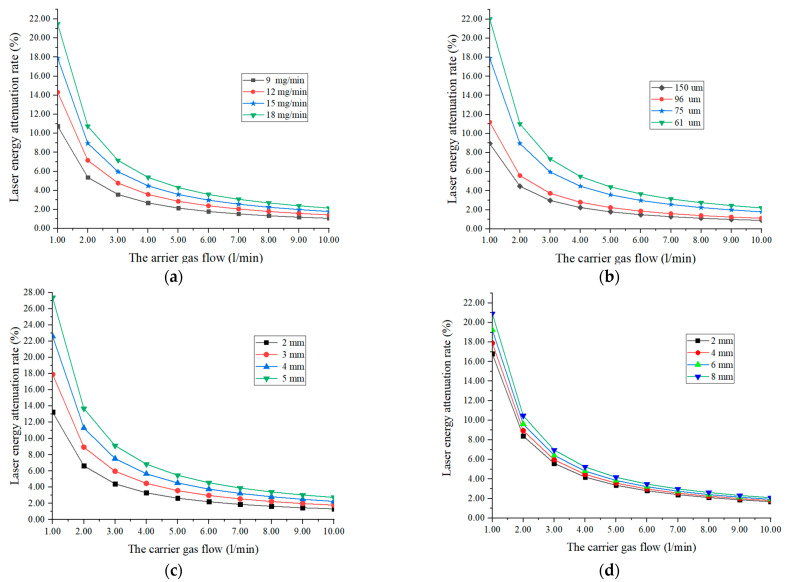
Curve of influence factors of laser energy attenuation rate: (**a**) The powder feeding speed $M_{p}$, (**b**) The powder particle diameter $\;r_{p}$, (**c**) Nozzle exit radius $\;r_{l}$, (**d**) Powder nozzle-to-center distance L, (**e**) The angle between the *x*-axis and the nozzle center *α*.

**Figure 3 sensors-22-02569-f003:**
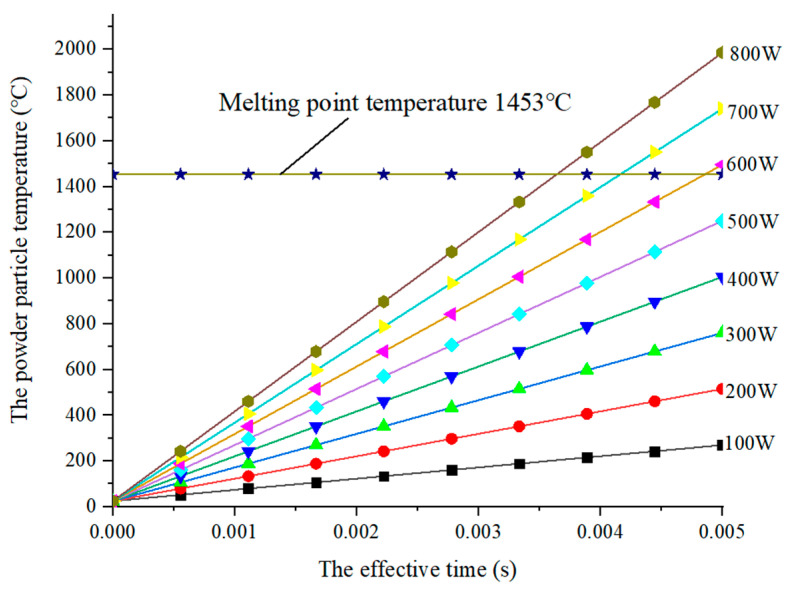
Single powder particle temperature change curve.

**Figure 4 sensors-22-02569-f004:**
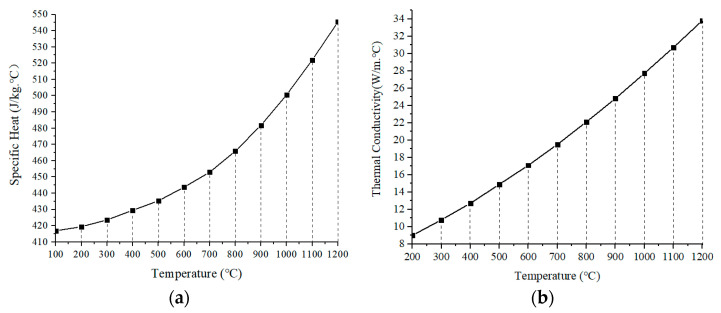
The thermophysical parameters of the material DD406: (**a**) Specific heat, (**b**) Thermal conductivity.

**Figure 5 sensors-22-02569-f005:**
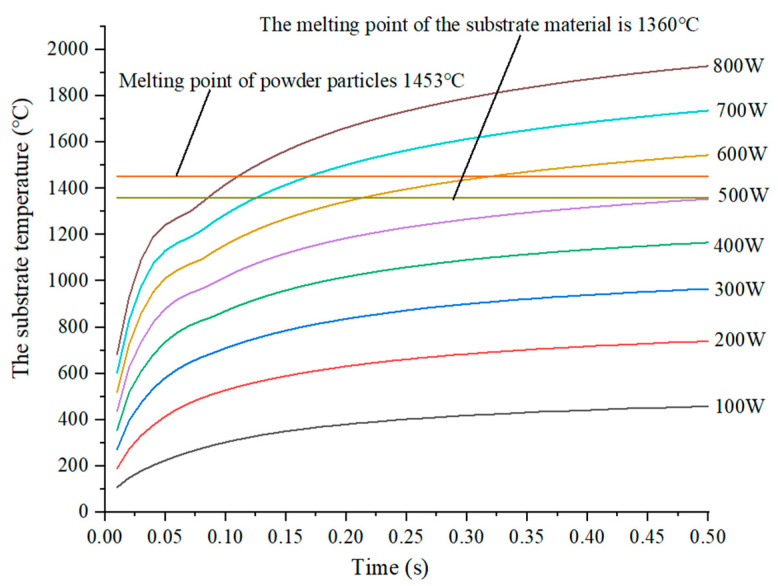
The relationship curve between the substrate temperature and the time.

**Figure 6 sensors-22-02569-f006:**
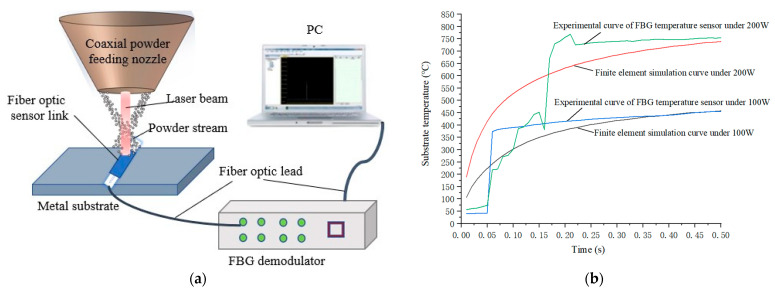
Temperature rise experiment verification diagram: (**a**) FBG sensor verification structure diagram of substrate temperature field, (**b**) Comparison of simulation curve and experimental curve of substrate temperature.

**Figure 7 sensors-22-02569-f007:**
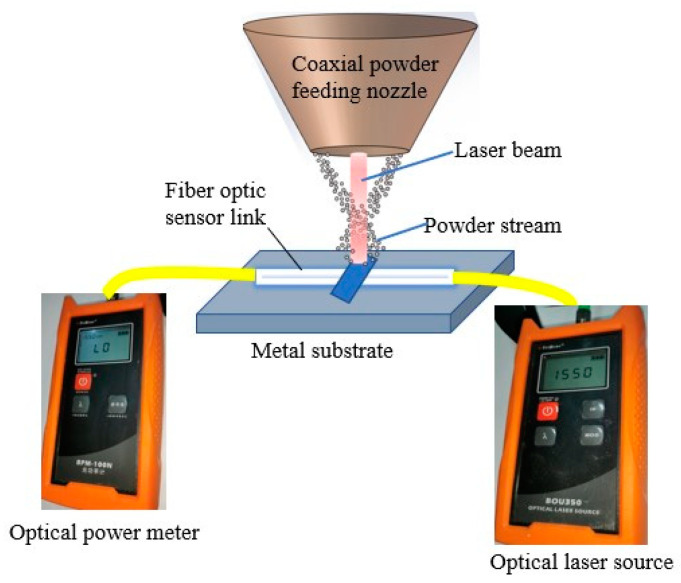
The structure diagram of the laser cladding experiment.

**Table 1 sensors-22-02569-t001:** Material properties used in the simulation.

Parameter	Unit	Value	Parameter	Unit	Value
Q	(L/min)	1~10	$r_{l\;}$	(mm)	2, 3, 4, 5
$M_{P\;}$	(mg/min)	9, 12, 15, 18	$W_{B}$	(mm)	3
L	(mm)	2, 4, 6, 8	*η*	(%)	35
$r_{p}\;$	(um)	61, 75, 96, 150	α	/	30°, 45°, 57.5°, 60°

**Table 2 sensors-22-02569-t002:** The selection of each parameter of the numerical analysis.

Parameter	Q (L/min)	$\;M_{P}\;\left(mg/min\right)$	L (mm)	$\;\;r_{p}\;\left(um\right)$	$r_{l\;\;}\left(mm\right)\;$	$\;\;\;W_{B}\;\left(mm\right)$	*η* (%)	α
Value	4	15	4	75	3	3	35	57.5°

**Table 3 sensors-22-02569-t003:** The selection of each parameter of the numerical analysis.

Power	Powder Feeding Speed	Power Loss	The Molten Pool
	12 g/min	0.01 db	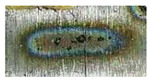
500 W	15 g/min	1.4 db	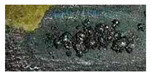
	18 g/min	5.36 db	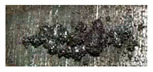
	12 g/min	1.2 db	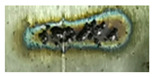
600 W	15 g/min	9.01 db	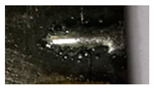
	18 g/min	24.47 db	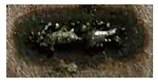
	12 g/min	7.77 db	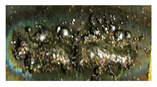
700 W	15 g/min	2.91 db	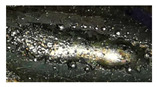
	18 g/min	34.63 db	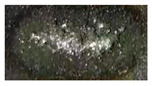
800 W	——	Fracture	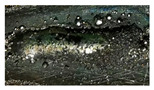

## Data Availability

Not applicable.

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
