# Peer review of "Modeling and Optimization of Laser Cladding Fixation Process for Optical Fiber Sensors in Harsh Environments"

_sensors, 2022, doi:10.3390/s22072569_

Round 1

Reviewer 1 Report

Modeling and optimization of laser cladding fixation process for optical fiber sensors in harsh environments 

  1. Check how the citation in the main text was written. The common style is LastName et al. (without the initial of the first name).
  2. Parameters/variables are encouraged to be put in nomenclature at the end of the manuscript.
  3. It is also mandatory that variables (Alphabet and Greek letters) use italic style. I found repeated misses in the manuscript.
  4. Pay attention to how the dimension and unit are written. Between them must be placed in a space. An example of the incorrect style is found in Table 1.
  5. T in the title of section 2.3 is in bold style. Is there a specific meaning to using this style?
  6. The mentioned problem in no. 4 is also found in Figure/Graphs.
  7. Make the writing style of graphs is uniform. Which one do you want to use? each first letter is capital or only the first letter in the first word is capital? the style is mixed in the current work and it is found repeated.
  8.  The experimental design has not clarified the details numbers of "d".
  9.  Every instrument needs to be clarified its origin, i.e., series, manufacturer name, city, country.
  10. Section 4.1 must be re-arranged. It is not acceptable that figures are placed right after the section title.
  11. Why style of Table 4 is different compared to others?
  12. Problems no 4 and 6 are found repeatedly in the Discussion.
  13. What kind of software was used in the simulation? Even though this method is used to produce main data (comparison with the experiment in Discussion), I do not see any details, including references and algorithms of the deployed software for simulation.
  14. Add recommendations for future works based on your findings.

Reviewer 2 Report

Dear Authors

The article concerns an alternative method of mounting fiber optic sensors to metal elements instead of traditional gluing. Due to their low shear strength, the adhesives do not perform well in difficult exploitation conditions. The authors proposed the use of laser cladding to attach fiber optic sensors. The article presents a simulation of the cladding process.  The influence of the assumed cladded parameters on the temperature distribution of the additional material and the substrate was assessed. The purpose of this was to carry out the cladded in such a way as not to exceed the melting point of the material from which the optical fibers are made. The obtained simulation results were verified on the basis of empirical research. Assumptions and simulation were performed correctly. The conclusions formulated in the article are only of a general nature and to a small extent refer to the obtained research results.

Detailed comments:

The additional material (powder) does not suppress the power emitted by the laser. It does not affect the laser power. Only the power of the electromagnetic wave beam is reduced.

The technological parameter of laser cladding is not the power of the laser used, but the power density expressed in W / m2

According to the SI, we denote liter as l

In the article, I miss the specification of the type of base material and the powder used for surfacing.

Round 2

Reviewer 1 Report

All comments have been addressed.